# Drilling of Cross-Ply UHMWPE Laminates: A Study on the Effects of the Tool Geometry and Cutting Parameters on the Integrity of Components

**DOI:** 10.3390/polym15193882

**Published:** 2023-09-25

**Authors:** Antonio Díaz-Álvarez, Marcos Rodríguez-Millán, Ignacio Rubio, Daekyum Kim, José Díaz-Álvarez

**Affiliations:** 1Department of Mechanical Engineering, University Carlos III of Madrid, Avda de la Universidad 30, 28911 Leganés, Spain; andiaza@ing.uc3m.es (A.D.-Á.); mrmillan@ing.uc3m.es (M.R.-M.); igrubiod@pa.uc3m.es (I.R.); 2School of Smart Mobility, Korea University, 145 Anam-ro, Seongbuk-gu, Seoul 02841, Republic of Korea; daekyum@korea.ac.kr; 3Institute of Innovation in Sustainable Engineering (IISE), College of Science and Engineering, University of Derby, Derby DE22 1GB, UK

**Keywords:** UHMWPE, machinability, drilling, damage

## Abstract

Ultrahigh-molecular-weight polyethylene (UHMWPE) is used in the defence industry mainly owing to its properties, such as excellent dimensional stability, excellent ballistic performance, and light weight. Although UHMWPE laminates are generally studied under impact loads, it is crucial to understand better the optimal machining conditions for assembling auxiliary structures in combat helmets or armour. This work analyses the machinability of UHMWPE laminates by drilling. The workpiece material has been manufactured through hot-pressing technology and subjected to drilling tests. High-speed steel (HSS) twist drills with two different point angles and a brad and spur drill that is 6 mm in diameter have been used for this study. Cutting forces, failure, and main damage modes are analysed, making it possible to extract relevant information for the industry. The main conclusion is that the drill with a smaller point angle has a better cutting force performance and less delamination at the exit zone (5.4 mm at a 60 m/min cutting speed and a 0.05 mm/rev feed) in the samples. This value represents a 46% improvement over the best result obtained in terms of delamination at the exit when using the tool with the larger point angle. However, the brad and spur drill revealed a post-drilling appearance with high fuzzing and delamination.

## 1. Introduction

Composite materials, in general, are increasing their presence in the vast majority of industries daily, especially in the aeronautical and automotive industries [1]. Although the parts manufactured with these materials are made near net shapes, auxiliary operations are necessary to achieve dimensional tolerances or make joints between different elements [2]. In this sense, the integrity of the composite material element depends mainly on its joints and not so much on its structure [3]. Mechanical joints (pins, bolts, rivets, screws, etc.) are the most used to assemble elements made of composite materials compared to adhesives [4,5].

However, to make mechanical joints, it is necessary to carry out some previous machining operations, such as dotting, drilling, and deburring [6]. During the machining of composite materials, there is a rapid generation of tool wear, mainly due to the nature of the material, normally made up of at least two different phases with highly abrasive fibres, which decrease its machinability [7]. The drilling of composite materials results in significant costs for the industry due to the damage that is caused, including the need for additional processes to achieve the required tolerances (30% of the production costs of the manufacturing industry are because of deburring processes [8]) and the rejection of pieces that do not meet quality standards [9]. The damage present during the machining and processing of composite materials is mainly related to inadequate cutting parameters or tool geometries, as well as excessive wear on the tool [10,11]. Many authors refer to the drill geometry as one of the most influential parameters in the drilling process. Different drills and tip angles have been analysed on composite materials [12,13,14,15]. Although the damage produced during the drilling of this type of material is always present, the damage mode is not always the same. Different damage modes are found in the literature, mainly including the delamination fundamentally in carbon fibre composite materials (CFRPs) [16,17], the fraying in fully biodegradable composites [18,19], and the fuzzing in aramid composites [2].

Polymeric composites have been gaining importance since their beginnings in the 1960s, standing out, in recent days, UHMWPE, above all, in their use as ballistic protection materials thanks mainly to their excellent strength-to-weight ratio and, therefore, excellent impact resistance [20]. Aside from its application in the military sector, UHMWPE is notable in bioengineering, where it is utilised as an artificial joint replacement material due to its biocompatibility, chemical stability, and low coefficient of friction among other properties [21]. However, it wears easily in this application compared to other harder materials and, thus, requires subsequent treatments.

Currently, the UHMWPE material is presented in industries in several grades, mainly owing to its different particle sizes and molecular weights [22]. Advances in manufacturing techniques (injection moulding, extrusion moulding, sintering, etc. [23]) for thermoplastic materials, such as UHMWPE, have made possible this variety in the particle size distribution, which allow components to be manufactured by focusing on the specifications of the material’s applications. UHMWPE materials exhibit 2.6 GPa and 87 GPa of tensile strength and modulus, respectively, with fibres that are up to twenty times stiffer and approximately four times stronger than nylon 6.6 fibres [24].

The drilling of UHMWPE composites is characterised by burr formation as the main damage mode due to the uncut fibres that appear both at the entrance and exit of the drill [25]. Both the viscoelastic behaviour of the material and the plastic deformation generated during machining make it difficult to select the proper cutting conditions and cause damage in the material, such as burrs and rough surfaces [26]. Thus, an adequate choice of tool geometries and cutting conditions is essential to minimise the damage generated during the drilling of UHMWPE composites. However, very few articles analyse the drilling of UHMWPE composite materials, making it necessary to conduct an in-depth and current study on this important topic.

Therefore, the following paragraph analyses the main contributions to date in the literature for UHMWPE materials manufactured through different processes and, therefore, having different properties.

Altan et al. [25] analysed burr damage and surface roughness generation during the drilling of UHMWPE materials in blocks. They used 8 mm diameter HSS twist drills uncoated and coated with TiN and TiCNfor the tests. Feeds of 0.006, 0.0125, and 0.025 mm/rev and cutting speeds of 30, 40, and 50 m/min were selected as the cutting parameters of the drilling tests. By means of the Taguchi method, they found that 0.025 mm/rev, 30 m/min, and the uncoated HSS drills were the optimal parameters to reduce burr generation. On the contrary, the coated HSS drills generate more heat due to their lower heat transfer coefficient and, therefore, more burrs at the exit. The influence of the cutting speed on the generation of burrs is almost entirely neglected compared with the feed and the tool type. Furthermore, the chip analysis shows higher deformation rates with coated HSS drills and low feeds. They did not find a clear trend between the selected cutting parameters and the surface roughness. Campos Rubio et al. [27] carried out drilling tests on UHMWPE polyacetal or POM and polytetrafluoroethylene (PTFE) billets by analysing the circularity error, the surface roughness, and the thrust force. HSS drills that were 10 mm in diameter, had a 25° helix angle, and had different point angles of 85°, 100°, and 130° were used. They established feeds of 0.05 and 0.15 mm/rev, whereas the spindle speeds were 1000, 3000, and 5000 rpm. They found that the circularity error decreases with higher feeds and point angles and the lowest spindle speed. The lowest spindle speed minimises the roughness, whereas combining the feed and the point angle enabled the achievement of better roughness. Moreover, the thrust force was minimised at a higher spindle speed and the lowest feed rate and drill point angle. Compared with other materials, UHMWPE exhibited the highest values of circularity and roughness and the lowest thrust forces of all the composites that were analysed. Ferreira Lizaro et al. [28] analysed the thrust force and the diameter variation when drilling UHMWPE-PE and PTFE composite materials. HSS twist drills that were 10 mm in diameter and had three different point angles (85°, 100°, and 130°) were used. For the cutting parameters, they selected 0.05 and 0.15 mm/rev for the feed and 1000, 3000, and 5000 rpm for the cutting speed. They observed that increments in the feed rate increased the thrust force under each test condition, highlighting that the drill diameter and feed rate are the most influential factors. In general, for most materials, the thrust force increases with the spindle speed, except for UHMWPE-PE (at 0.05 mm/rev and a 100° point angle), owing to the increase in the softening of the material by means of increments in the temperature with increasing spindle speed. Regarding the diameter variation, they showed that the error decreased with increments in the point angle. 

Therefore, because the drilling process of the UHMWPE composite has not been analysed adequately for materials manufactured using hot-pressing technology, which is a common method for manufacturing protective elements against impacts [29,30], it is necessary to carry out a more in-depth study. Thus, this paper focuses on the study of the drilling process of UHMWPE composite materials manufactured by means of hot-pressing technology. The different cutting parameters and their influence during the process were analysed during the drilling. Three cutting speeds of 30, 60, and 90 m/min and feed rates of 0.025, 0.05, and 0.15 mm/rev were tested. Moreover, the drill geometry and its impact on damage generation were analysed with two different HSS drill geometries and a brad and spur drill (no studies have been found to date for this last drill geometry). The influence of the drill point angle was studied through two different point angles of 118° and 80° in twist drills. During the study, the cutting thrust force and the damage generation were quantified and related to the main cutting parameters and selected geometries. Pearson correlation analysis was carried out to study the impact of the different cutting parameters on the generation of cutting forces and the main damage that was observed. The dominant damage observed during the tests was delamination, which is highly dependent on the cutting speed and the feed but significantly on the cutting speed. In addition to the cutting speed, we verified that the point angle of the drill is a crucial factor in the development of delaminations when drilling UHMWPE composites. Among the drills that were tested, the 80° twist drills at a cutting speed of 60 m/min and a feed of 0.05 mm/rev achieved the lowest delamination value.

## 2. Experimental Section

This section will describe the material that was used to carry out the tests and the main equipment in relation to the machines, drill bits, and cutting conditions. In addition, the main damage modes found during the drilling of the UHMWPE composite material will be described.

### 2.1. Workpiece Material

Ultrahigh-molecular-weight polyethylene fibre composites were used as the workpiece material. Owing to its good mechanical properties, light weight, and excellent impact resistance, UHMWPE is widely used in applications, such as armour or personal protection equipment, anchor ropes, and fishing nets. The UHMWPE laminate’s properties are summarised in Table 1. The laminate was manufactured using hot-pressing technology, following a procedure similar to that presented by Haris and Tan [31]. The thickness of the specimen was 9 mm, which is considered as the standard for combat helmets, and the laminate density was 0.97 g/cm^3^.

### 2.2. Drilling Cutting Tests

A machining centre (B500 KONDIA, Kondia, Elgoibar, Spain) equipped with a dynamometer (Kistler 9123C, Winterthur, Switzerland) to quantify the thrust force (from −2 to 2 KN) and torque (from −20 to 20 Nm) was used for developing the drilling tests (Figure 1). The tests were carried out without a coolant to preserve the material’s mechanical properties.

Two different high-speed steel drill geometries that were 6 mm in diameter were used for developing the drilling tests (Figure 2); both types of drill bits were supplied by Guhring [32,33]:HSS twist drills that had 118° and 80° point angles. They are widely reported in the literature owing to their high industry availability and medium cost. These drill geometries are the most used during the analysis of the drilling of the main composite materials analysed in the literature and, especially, the UHMWPE composite [25,27,28]. Thus, the 118° drill will be taken as the reference for this study (the tool with the smaller point angle was customised to maintain the same rake angle and clearance angle as those of the drill bit with the larger point angle).HSS brad and spur with a 110° point angle (denoted). Manufacturers have especially recommended this drill geometry to eliminate burrs and delamination generated during the drilling of composite materials. Several authors have analysed its geometry during the drilling of composite materials, but it has not been reported for the UHMWPE composite [34,35,36].

**Figure 2 polymers-15-03882-f002:**
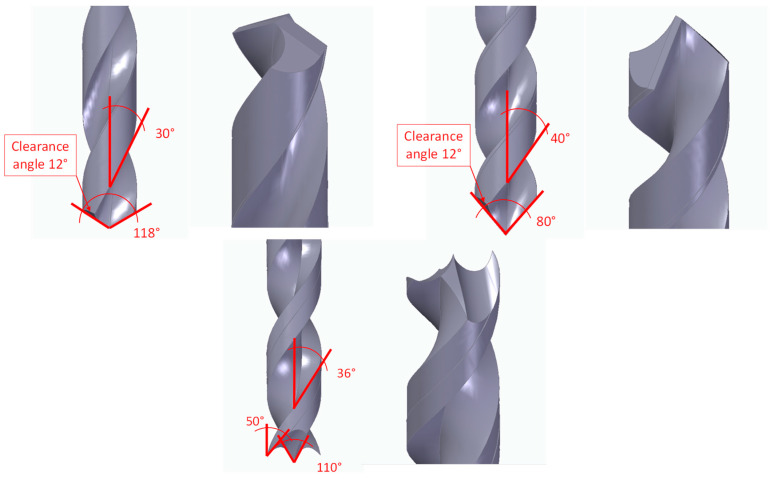
Different HSS drill geometries tested during the drilling of the UHMWPE composite.

For the development of the drilling test, three different cutting speeds and feeds were established by considering a combination of factors, including the recommendations provided by tool manufacturers, insights from material suppliers, and findings reported in the literature by other authors. Therefore, the main cutting parameters analysed in this study are summarised in Table 2.

### 2.3. Damage Modes

During the development of the drilling tests, delamination, burrs, and fuzzing were the main damage modes in the processing of the UHMWPE composite material. This damage was determined visually by means of the calculation of the maximum diameter within the damage that appears throughout the images that were obtained using an optical microscope (Optika SZR, Ponteranica, Italy). Delamination appears during the processing of the material, both at the entrance and at the exit, due to the pull out of the layers during the entry of the drill by means of its cutting edges or as a response to the push out of the tool in the last layers at the exit. Therefore, this leads to an increase in the thickness of the workpiece in the affected area and a reduction in its properties due to the separation of the material layers (Figure 3a).

Ghabezi et al. [37,38] already distinguished between burrs and delamination, highlighting two types of delamination based on the area where they appeared during the drilling of composite sandwiches. Thus, they pointed out one damage mode as fibres around the hole, and the other damage mode was the existing uncut fibres and matrix that appeared in the hole.

The W_d_ factor (workpiece damage) is used to quantify the workpiece damage. Wd is defined as the maximum diameter that includes the whole damaged area, including all the delamination, burrs, and fuzzing (D_max_); and the nominal diameter (D) refers to the drill diameter. Several authors have used this factor to quantify the damage generated during the drilling of composite materials [2,39]. 

During machining, burr generation is another critical phenomenon that is observed. Therefore, burrs have been generated at the hole edge during the processing of plastically deformed materials [40,41]. Regardless of whether the part’s material is a metal or composite, burrs are undesirable elements that limit the surface quality of the part and can cause problems during its assembly with other elements to set components [42]. Moreover, the piece’s performance could be at risk of causing the blockage of gases or liquids when the element its used to channel them. Thus, postprocessing is sometimes required to achieve the desired dimensional quality, which increases the manufacturing time and cost. Xu et al. [43] highlighted during the drilling of woven GFRP composites that the burr phenomenon observed during machining was attributed mainly to the bouncing-back effects and mechanical recession of fibres under the bending loads of the tool lips.

Burrs are measured through the optical images of holes. Using the plane that defines the surface of the test piece as a reference, the height of the burrs was calculated at both the entrance and exit points.

Therefore, in the present study, both the delamination and the burrs generated during machining were quantified at the entrance and at the exit of the hole.

## 3. Results and Discussion

In this section, the main results obtained during the drilling of the UHMWPE composite material are analysed in terms of forces and damages induced during processing. Moreover, a statistical method is used to analyse the obtained results.

### 3.1. Cutting Force Analysis

The thrust force and torque values were recorded and analysed for each test. Figure 4, Figure 5 and Figure 6 illustrate the variation in the thrust force and torque across the thickness of the sample during the drilling process, using both cutting geometries, at a cutting speed of 30 m/min and a feed rate of 0.025 mm/rev.

For the 118° twist drill geometry (see Figure 4), the thrust force is the maximum when the entire conical shape of the drill has entered the specimen (point B). Then, it decreases slightly along the thickness of the sample with the machining of the different layers until the bottom surface is reached (position C). In the case of the torque evolution, the values obtained during drilling progress until reaching the maximum at position C, when the drill comes to the last layers of the sample. The drill exits the material between positions C and D, which is associated with reducing the force to reach a valley.

In the case of the 80° twist drill geometry (see Figure 5), both the thrust force and torque follow the same tendencies as those obtained for the 118° point angle but with slight differences. Thus, the maximum value reached for the thrust force was less than that obtained for the 118° point angle for all the equivalent tests, mainly owing to its sharper geometry. Nevertheless, in the case of the torque, the maximum value that was reached was more significant for the 80° twist drill geometry compared to the 118° twist drill geometry. Furthermore, these results also emphasise that as the length of the cutting edge in contact with the material is greater for the tool with the 80° point angle compared to the tool with the 118° point angle, the average stresses on the cutting edge will also be lower for the former tool than for the latter tool.

For the brad and spur drill (see Figure 6), the thrust force increased until point D, when the outside cutting edges of the drill reached the sample material. From point D, the thrust force slightly increased then decreased at point E owing to a reduction in the material strength by means of the drill approaching the bottom surface of the material. Concerning the torque progression, no clear trend could be observed for the drill movement through the thickness of the specimen.

The different cutting parameters, both the cutting speed and feed, and their impacts on the generation of thrust force and torque were analysed for each established drill geometry. Thus, in Figure 7, the maximum thrust force and torque values are represented for each tested cutting condition. The following analysis will be based on the 118° twist drill as the reference tool.

For the 118° twist drill geometry, the trends in the thrust forces are easily observable. As the feed increases, the magnitudes of the cutting forces also increase, mainly owing to an increment in the undeformed chip cross-section. On the other hand, as the cutting speed increases, the magnitudes of the thrust forces decrease. This relation can be explained by the temperature that was reached as the cutting speed increased. The temperature reached by the specimen will be higher because the heat generated during the cut has less time to dissipate from the cutting area. It will cause material softening, which requires less force for machining. Temperatures above 70 °C may be reached, which reduce the UHMWPE resistance, as previously reported in the literature [44].

Despite differences in the manufacturing of the test material and potential variations in the properties of UHMWPE, these findings align with those reported by other researchers who have drilled similar pieces of this material [25,28]. Similar trends have been observed regarding torque, wherein an increase in the undeformed chip section leads to an elevation in torque. Moreover, a decrease in torque is also observed with increasing cutting speed, which is justified by a greater softening of the material.

Because of the results obtained in terms of the forces and defects found in the holes drilled with the 118° twist drill geometry, an attempt was made to minimise the forces (because this defect is the most difficult to repair in a possible postprocessing) using a tool with a smaller point angle. Another point angle was used to try to minimise the thrust force, which seems to be the most important factor that governs the damage in other composite materials [2].

Figure 7 compares the forces and torques obtained with the 118° and 80° twist drill geometries. The same trends obtained for the 118° twist drill were observed again for the 80° twist drill. Thus, the forces and torques both increased and decreased as the feed and cutting speed increased, respectively. These results agree with those obtained by other authors in drilling pieces of the UHMWPE material [25,28]. However, as expected for the 80° twist drill, the obtained thrust force values were significantly lower, up to 25% in the worst case, while the average torque values were 14% higher under the tested conditions. In the case of the 80° twist drill, as shown in Figure 8a, the chip is thinner, and the length of the edge in contact with the material is greater than that of the 118° tool. This causes two effects to appear: on the one hand, there is greater friction between the tool’s edge and the material, which produces more significant heating of the material due to friction. Therefore, it causes the temperatures reached by the material under the tip of the drill to increase to a greater extent than that for the 118° twist drill, which softens the material; therefore, less thrust force is required to remove the material than that required with the 118° twist drill geometry. On the other hand, the radius of the sharp edge of the tool has a greater effect on the cut, producing an increase in the torque because the effective rake angle is smaller. Figure 8b shows the effect of the cutting edge’s tip radius with respect to the feed. 

As stated in Section 2.1, the brad and spur drill was developed to carry out drilling processes in composite materials with a polymeric matrix and a continuous carbon fibre reinforcement [2] to improve hole quality by minimising the number of uncut fibres and delamination that are generated.

In Figure 7, the thrust forces and torques of the brad and spur drill with respect to the 118° and 80° twist drill geometries are represented. For this drill, a different trend appeared in the thrust forces that had not been observed for the rest of the tested tools. The first significant observed effect is that for feeds of 0.025 and 0.05 mm/rev, the thrust forces generated by the brad and spur drill are greater than those obtained for the 118° twist drill and lower in the case of the 0.15 mm/rev feed.

Concerning the torque, for all the cases, lower values are observed than those obtained for the 118° twist drill. In the specific case of the brad and spur drill, a different behaviour was noted concerning the impact of the feed on the thrust force. Increasing the feed per revolution did not increase the thrust force owing to the larger chip section. In fact, when feeds ranged from 0.05 to 0.15 mm/rev, the thrust force was reduced across all the tested speeds. The reason lies in the complexity of the tool’s edge. This means that the chip exit is not as direct as those for the other two tested geometries. In this case, unlike when this geometry is used for machining thermosets, the fibres have excellent flexibility and, at the same time, high strength, on the order of 7 GPa, and a modulus of 235–325 GPa [45]. These features show that the material tends to adhere at the tip of the drill, even melting, to generate a stopper, like the one shown in Figure 9, for feeds of 0.025 and 0.05 mm/rev. Owing to this adhered material, the recorded thrust force was higher for the lower feeds (0.025 and 0.05 mm/rev). Thus, authors have already reported a similar fact during the machining of aramid composites with the same geometry and low feed rates [2].

Regarding the effect of the cutting speed, as expected, the general tendency is that the thrust force reduces as the speed increases. This is insignificant for the 0.15 mm/rev feed, where the cutting speed does not seem to affect the thrust force. This phenomenon can be attributed to thermal softening, whereby feeds of 0.025 and 0.05 mm/rev cause the material to accumulate on the drill, preventing the proper cutting of the edges and filling the concave part of the tool with the material. As a result, the temperature distribution on the drill face becomes more akin to that on the face of a 118° twist drill. However, at a feed of 0.15 mm/rev, it is possible that the generated heat did not have the same exhaust lines owing to being trapped in the concave zone of the tool. Therefore, the temperatures reached at the front of the drill would be similar for all speeds and would generate similar thrust forces regardless of the cutting speed.

Regarding the torque results for the brad and spur drill, very similar, and almost independent, values can be observed for the value of the feed per revolution and the cutting speed, which is slightly higher for the highest feed (25.6% at 30 m/min). This suggests that the thickness of the undeformed chip analysed in the feeds has a negligible effect, considering that the undeformed chip is proportional to the feed (refer to Figure 6, detail C). This could be attributed to the peripheral edge and tip of the drill, which disrupt the continuity of various layers and allow the drill to only push the material near its edge and tip. Moreover, in the intermediate areas, the material is at a high temperature for the material in question, making it very soft and offering little resistance. In that area, the amount of resistance required to delaminate the material is very low. It is very likely that this confined material will delaminate during the feed movement and will not resist being removed from the base material.

### 3.2. Surface Damage

#### 3.2.1. Delamination Analysis

The delamination damage was quantified for each test, represented in Figure 10, for all the cutting parameters and drill geometries at the tool’s entrance and exit.

Starting with the 118° twist drill geometry and the tested conditions, delaminations were found at both the entrance and the exit, and it was necessary to distinguish those two areas during the analysis. 

In Figure 10, the delamination damage found at the entrance and exit of the hole are represented. For all the cases, higher delamination values were found at the exit of the hole than at the entrance, and the same behaviour was observed for other composite materials with a thermoplastic or bio matrix [2,46]. It was not possible to observe a clear effect of the feed on the delamination at the exit of the hole for the range of tested speeds. This was not expected because increasing the feed also increases the thrust force, which favours the separation of the last layers. However, authors have previously reported similar behaviours while machining aramid composites [2]. The effect of the cutting speed was very significant on the observed delaminations. Two groups could be clearly observed regarding delamination at the hole exit. 

For the two lowest tested cutting speeds (30 and 60 m/min), the highest delamination values were reached at the exit side, which were much higher than that in the case of the 90 m/min speed. At low speeds, a part of the heat generated by machining has time to reach areas far from the hole, which softens the material. This phenomenon and the greater thrust forces at lower speeds justify the finding of higher delamination values at cutting speeds of 30 and 60 m/min. 

Moreover, another very significant finding is that when the delamination damage value reached around 22 mm, it did not seem to be affected by the feed; the higher delamination damage value at a speed of 60 m/min indicates that it is the most unfavourable condition. Thus, the heat generated in the cut reaches areas away from the edge of the hole, softening the resin that holds the fibres together in addition to having to support thrust forces very similar to those generated at a speed of 30 m/min.

At 90 m/min, it can be understood that less generated heat reaches areas far from the hole, which means that the amount of stress that the joint between the layers can withstand without breaking is more significant. Moreover, this phenomenon is possible because there is no significant cumulative heat effect; therefore, the material embrittlement factor becomes very important, and increasing the strain rate makes the material less ductile, which favours a clean fibre cut [47].

Regarding the delamination at the entrance, it does not seem to be influenced much by the cutting speed because the heat accumulation due to the different number of revolutions at the front of the drill does not exist at the entrance of the hole, so the most important factor is the embrittlement of the material due to higher strain rate [47], which helps to localise the stresses closer to the hole border and, thus, minimises delamination. At the hole’s entry, the drill’s helix angle and the tool feed seem to be more critical. At speeds of 30 and 60 m/min, the delamination values clearly increase with increasing feeds; this effect was not found for the 90 m/min cutting speed, for which less ductile behaviour of the material is expected. For the 80° twist drill and tested conditions, like with the 118° twist drill geometry, the main defects were delaminations at the entrance and exit. These effects could generate a decrease in the mechanical properties of the component. Additionally, burrs at the entry and exit of the hole, which could make it difficult to fabricate joints, negligible fuzzing was appreciated (it is analysed in subsequent subsections).

In Figure 10, the delamination damage found at the entrance and exit of the hole for the 80° twist drill is represented. Like with the 118° twist drill, the delamination at the exit was greater than that found at the entry. Another important finding is that at equivalent cutting parameters under all the tested conditions, the delamination at the entry was lower for the 80° twist drill geometry than for the 118° twist drill geometry. Table 3 lists the relative reductions in the damage factor at the entrance and exit for the 80° twist drill compared to the 118° twist drill. This is directly related to the lower thickness of the undeformed chip obtained as the drill angle is decreased (see Figure 8a). This reduces the magnitudes of the required forces and stresses in the areas near the hole’s edge. 

With regard to delamination at the exit of the hole, two behaviours should be highlighted: at the lowest test speed, 30 m/min, the highest delamination values are reached, in the same order for both the 118° and 80° twist drill geometries. (In the best case, the delamination damage was around 22 mm.) This is because, at this low cutting speed, the heat generated by machining has time to reach areas that are very far from the edge of the hole and soften the material. This and the greater thrust forces cause the greatest delaminations. At lower feeds with the 80° twist drill, greater delaminations are reached at the exit, which could be attributed to greater buckling of the material due to the point angle of the drill, which would generate greater stresses in the material. 

When the cutting speed is increased, the heat has less time to escape from the cutting zone. On the other hand, the material hardens owing to the higher strain rate [47]. In the case of the sharpest tool, as shown in Figure 7, thrust force values similar to those obtained by the 118° twist drill at a cutting speed of 90 m/min were obtained for the cutting speed of 60 m/min. Therefore, it is not surprising that the delamination values at the exit for the 80° twist drill at the cutting speed of 60 m/min are even lower than those for the 118° twist drill at 90 m/min. 

At the cutting speed of 90 m/min, lower delamination values continue to be obtained at the exit for the 80° twist drill, mainly owing to the already mentioned lower thrust forces. Another finding to highlight is that very similar delamination values are obtained for this tool geometry regardless of the cutting speed in the range between 60 and 90 m/min and at the studied feed. Delamination at the entrance appears to be less affected by the temperature generated during cutting. An even smaller effect is expected compared with that generated by the 118° twist drill because less material is removed in the first revolutions of the tool when it starts touching the material’s surface. Hence, the most important factor is the embrittlement of the material due to the higher strain rate [42]. This helps to localise the stresses closer to the hole border, which minimises delamination and, in general, less delamination is generated as the cutting speed increases. 

Delamination defects were found in relation to the brad and spur drill and the tested conditions. Because several burr defects were present in the holes generated by the 118° and 80° twist drills, this tool was expected to generate almost negligible burr formation. However, the appearance of uncut fibres that could make the assembly of components complex is significant (as is analysed in subsequent subsections).

As previously mentioned in Section 2.2., this tool is designed mainly for composite materials with a more fragile thermosetting matrix than in the case of the studied material. However, its performance was examined to avoid burrs and delamination. It did not show a good behaviour at the lowest feeds, owing to the tendency of the material to become trapped at the tip of the tool, creating a plug that invalidated the possible beneficial effects of the geometry. 

In Table 3, the values are listed relative to the delaminations found for the brad and spur drill compared to the 118° and 80° twist drills under the different tested conditions. Based on Figure 11, it can be reasonably inferred that the geometry of the brad and spur drill results in less radial force in the cutting-edge area compared to the thrust force of the 118° twist drill. Therefore, in general, for all the tested cases, the delamination at the entrance is less severe with the brad and spur drill than with the 118° twist drill (with less apparent plug formation at the drill tip in the entrance area). However, at the highest cutting speed, the delamination at the entrance was higher than those found using 80° twist drill. 

In the case of the delamination at the exit, only the case of the 0.15 mm/rev feed at the different cutting speeds will be analysed because the tool does not cut properly through the formed plug, which makes this analysis meaningless. A clear trend was observed for the 0.15 mm/rev feed, where an increase in the cutting speed decreases delamination. Moreover, at all the cutting speeds, the delamination values were lower than those found for the 118° twist drill, and the lowest value was obtained at the 90 m/min cutting speed and 0.15 mm/rev feed (8.25 mm). Authors have already proposed the explanation for this in a previous work [2], in which the drill geometries and forces generated on both the sample and the tool were analysed (Figure 11).

Thus, the differences between the 118° twist drill and the brad and spur drill regarding the resulting forces show that the forces push the material outwards for the first geometry. In contrast, the forces push the material inwards for the second. Therefore, those resulting forces that push the material outwards favour material damage. 

In the case of the brad and spur drill, the decrease in delamination with increasing cutting speed is attributed to the material becoming more brittle due to higher strain rates. This means that the deformations do not propagate through the different layers when breaking with fewer deformations. However, as in the case of the delamination at the entry side, the best result was found using the 80° twist drill at the higher cutting speed. Therefore, the optimal approach to machining this type of material with regard to delamination is to use small cutting angles to minimise the thickness of the uncut cross-section of the chip. This directly results in less force near the hole’s edge and high speeds to increase the material’s brittleness.

The analysis of the delamination damage can also be visualised in a clearer 3D map for the three different investigated drill geometries and cutting conditions, as shown in Figure 12. The 3D map clearly shows that the 80° twist drill has the best performance in terms of minimising delamination damage, making it suitable for manual drilling where precise control over the feed and cutting speed may not be achievable.

The authors have tried to establish a correlation between the thrust force and the delamination factor at the entry and exit sides, as shown in Figure 13. However, a nonclear tendency can be highlighted. Apparently, as the thrust force increases, the delamination factor increases for both tested twist drills, but it is not fulfilled in all the cases. Hence, the thrust force cannot be taken as a variable to predict delaminations in the studied material. In the case of the brad and spur drill, as previously mentioned, owing to the significant problem with the adhesion of the material on the tip of the tool, which directly affects the thrust force, a non-reliable correlation can be established.

#### 3.2.2. Burr Analysis

The appearance of burrs in this type of material when it is manufactured by extrusion is very normal owing to its very ductile behaviour [25]. The material used for this study was manufactured through laminates stacked together using the hot-pressing technology, which enables the apparition of fuzzing, a more common defect that appears in composite materials during drilling operations [2]. Figure 14 presents the burr values for both the 118° and 80° twist drills and fuzzing for the brad and spur drill at the entrance and exit of the hole. 

In Figure 14, for the 118° twist drill, a trend could be found that was similar to those obtained for the forces. Everything seems to indicate that as the feed increases, the number of burrs generated at the exit side increases, in general, at all the tested cutting speeds, which is expected because this is related to the cross-section of the undeformed chip. Thus, as the cross-section of the chip increases, the level of burrs increases because more material is deformed per revolution of the drill bit [25]. The most significant effect of the feed is found at a cutting speed of 60 m/min because when the greatest delaminations are caused, the structure is not so rigid, causing the material to flex under the drill’s action and making a clean cut of the drill impossible. 

Flexion will increase proportionally with the thrust force, which increases with the feed. In the case of a cutting speed of 90 m/min, although the area through which the drill moves forwards must be hotter, in view of the results of the thrust force, on the periphery of the hole, the number of delaminations is not as significant as in the case of the 60 m/min cutting speed. This, together with even higher strain rates, means that the material is not as ductile in the periphery as in more central areas of the hole, which means that the tool can cut the hole more cleanly without leaving so many burrs [47]. 

In the case of the burrs at the entrance, note that as in the case of delamination, this does not seem to be influenced much by the cutting speed. The effect of successive passes on the accumulation of heat, which could soften the material at the edge of the hole, causing it to soften, is not so marked. Also, the generated heat is more easily absorbed more quickly by convection and conduction in the area than at the bottom of the hole. However, it does seem to be more sensitive to the feed value of the drill, mainly owing to the larger chip section, as authors have previously demonstrated [42].

The 80° twist drill geometry did not show a significant reduction in the level of burrs at the entrance and exit, under any of the tested conditions, as observed in Figure 13. Furthermore, the level of burrs found was consistently greater than or of the same order at both the entrance and exit, across all the cases. 

In the case of the burrs at the entrance under the tested conditions, a significant effect is not appreciated for the tool’s geometry. However, for the burrs at the exit, a considerable influence is appreciated for this geometric factor. The sharper geometry made the material try to flow more in the radial direction in the areas at the edge of the hole. This was caused by the higher component of the forces in the radial direction and the greater bending of the material when the tip penetrated the thickness of the specimen. The effect of the feed can be clearly observed at higher feeds. Thus, at the same speed, more burrs were generated at the exit (the trend of the levels of burrs generated at the entrance is not very evident with this variable) owing to a larger chip section (this behaviour is identical to that observed with the 118° twist drill). 

Regarding the cutting speed, this does not seem to significantly influence the level of burrs generated at the entrance. However, similar to the trend that was detected for the delaminations (the authors justify this finding with the same argument as that presented for the 118° twist drill), it seems to influence the number of burrs produced at the exit, and the most burrs were generated at a cutting speed of 60 m/min. Finding more burrs at a cutting speed of 60 m/min instead of 30 m/min does not seem strange because, at higher speeds, the highest temperatures of the material are expected to be found in the area of the hole. As the temperature reached by the material increases, the material softens, making it able to withstand greater deformations caused by the cut without breaking, which is what happened with the 118° twist drill. However, what does not seem so obvious is that at a cutting speed of 90 m/min, more burrs are not produced. In contrast, in the case of the 118° twist drill, this is justified by two reasons.

On the one hand, although significantly more delaminations were found at a cutting speed of 60 m/min than at 90 m/min, the material of the test piece would flex more, making the material more flexible and, therefore, more difficult to cut cleanly. On the other hand, the higher cutting speed favoured hardening through strain, causing the area where the burrs were formed to behave more brittle compared to that at a speed of 60 m/min. Although the 80° twist drill generates more material strength, the main reason for fewer burrs at 90 m/min is the higher material embrittlement and strain rates [42,47].

For the brad and spur drill, no significant burrs were found, as shown in Figure 15, and mainly uncut fibres were found. This defect is widespread in composite materials with ductile fibres, as in the case of biodegradable composite materials [46].

As in the case of the study of the delaminations, for feed rates of 0.025 and 0.5 mm/rev, conclusions can only be drawn from the fuzzing at the entrance when there is still no material adhered on the surface of the cutting edge, and the tool cuts appropriately. Therefore, there is a clear trend for an increase in fuzzing at the entrance at higher feed rates. However, the cutting speed did not significantly affect the fuzzing appearance. As for the fuzzing at the exit, it was noticed that at a feed rate of 0.15 mm/rev, the values were comparable to those obtained at the entrance, with a slight reduction in fuzzing observed at higher cutting speeds. This is probably because the higher strain rate resulted in greater embrittlement of the fibres. This trend can be observed in Figure 15. 

### 3.3. Pearson Correlation Analysis

During the analysis of the results, a Pearson correlation coefficient was used, also known as Pearson’s r, which is a statistical measure that indicates the strength of the linear relationship between two continuous variables. The input variables are the cutting speed and feed rate. The output parameters are the thrust force, torque, level of burrs at the drill’s entry, number of burrs at the drill’s exit, delamination at the entrance, and delamination at the exit. The Pearson correlation coefficient is calculated by dividing the covariance of the two variables by the product of their standard deviations. Its value can be anywhere from −1 to 1, with −1 meaning a perfectly negative correlation and 1 meaning a perfectly positive correlation.

As indicated in the following paragraphs, Pearson’s analysis agrees with most conclusions from the previous authors’ findings. 

The correlations are displayed in Table 4. The correlations between the cutting speed and output variables are negative; however, the correlations between the feeding rate and the rest of the variables are positive, except for the correlation between the feeding rate and exit delamination, which is negative. There is a strong link between the cutting speed and delamination, and the links between the feed rate, torque, and burr entry are not as strong.

Table 5 lists the relationships between the drilling and output parameters for the three studied drills. For the cutting speed, the 118° twist drill has a medium correlation with the torque and delamination at the exit. However, the 80° twist drill shows a strong correlation with the delamination at the exit and a medium correlation with the torque and level of burrs at the exit. The brad and spur drill, which is the drill with the most different geometry, only has a strong correlation with the exit fuzzing and a moderate correlation with the thrust force. The feed rate parameter shows, in general, an excellent correlation. For the 118° twist drill, there is a strong correlation with the thrust force, torque, level of burrs, and delamination at the entrance. However, the feed rate strongly correlates with the thrust force and torque for the 80° twist bit. The brad and spur drill strongly correlates with the thrust force and entry fuzzing.

## 4. Conclusions

In this work, three different drill geometries were tested for machining the UHMWPE composite at three cutting speeds and feeds to minimise the main defects/damage generated by machining, which are mainly delaminations, burrs, and fuzzing for this material. The main conclusions of this study are summarised below:The delaminations that were found were highly dependent on the cutting speed and the feed but very significantly on the cutting speed. Clearly, for all the tested tools, the number of delaminations observed in some cutting regimes dropped drastically.In addition to the cutting speed, the point angle of the drill was identified as a crucial factor in the generation of delaminations during the drilling of UHMWPE.The 80° twist drill achieved the lowest delamination value among the tested tools at a cutting speed of 60 m/min and a 0.05 mm/rev feed (5.4 mm). It was significantly lower than the lowest delamination value obtained with the 118° twist drill (10 mm in the case of the 90 m/min cutting speed and 0.025 mm/rev feed).For more productive parameters, both the cutting speed and feed should be increased, and it is possible to verify that the drill that minimises delamination is the one with the lowest point angle (the 80° twist drill), which generated a delamination of 7.75 mm; the increase in delamination is 43.51% with respect to the minimum value obtained under the optimal conditions. In terms of the cutting force, again, the drill with the lowest point angle (the 80° twist drill) obtained the best results when machining at the highest cutting speed and the lowest feed.Regarding the level of burrs found, it should be noted that these were only produced with the 118° and 80° twist drills because with the brad and spur geometry, no burrs were found, only fuzzing.The 118° twist drill was the best tool for minimising the level of burrs, as observed in the results obtained at the highest cutting speed (with no significant effect of the feed rate for the highest cutting speed that was tested).It is worth pointing out that although the brad and spur drill is used to reduce the level of burrs, it leads to more fuzzing and delamination compared to the tool that performs the best in these aspects.Based on the findings related to forces and damages, it can be concluded that the tool with the lowest point angle is the most versatile because less delamination was found and because burrs could be removed after postprocessing.

## Figures and Tables

**Figure 1 polymers-15-03882-f001:**
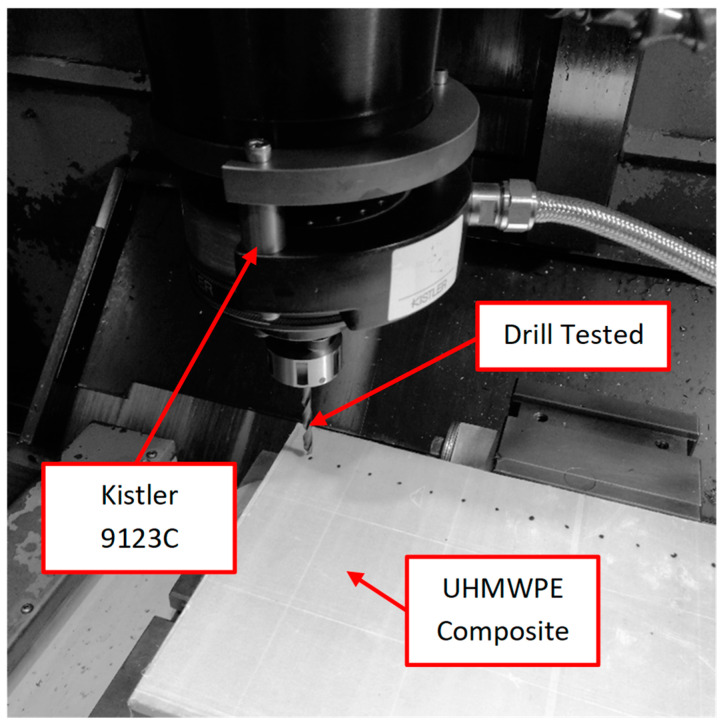
Machining centre (B500 KONDIA) equipped with a dynamometer (Kistler 9123C) for the drilling tests.

**Figure 3 polymers-15-03882-f003:**
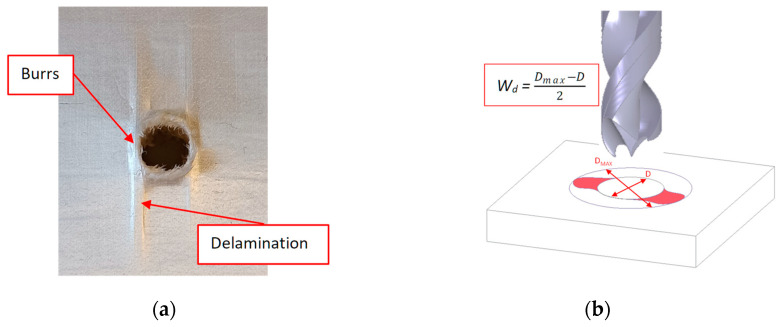
(**a**) Delamination at the entry during drilling of the UHMWPE composite with a 118° twist drill at a 30 m/min cutting speed and a feed of 0.15 mm/rev; (**b**) measurement of the damage derived from processing.

**Figure 4 polymers-15-03882-f004:**
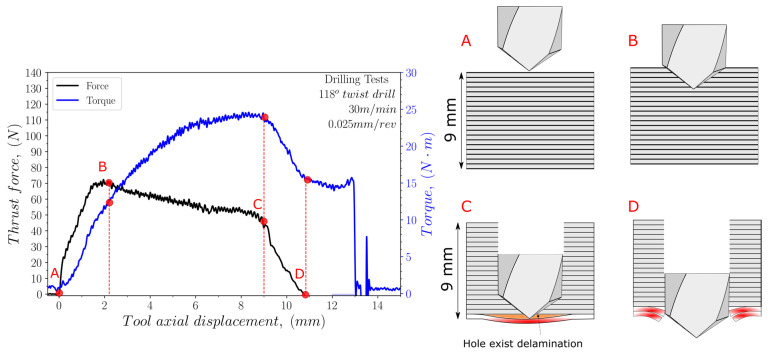
Evolution of both the thrust force and torque during the drilling of the UHMWPE composite with the 118° twist drill geometry at a cutting speed 30 m/min and a feed 0.025 mm/rev (A–D letter correspond to different instants during the drilling operation).

**Figure 5 polymers-15-03882-f005:**
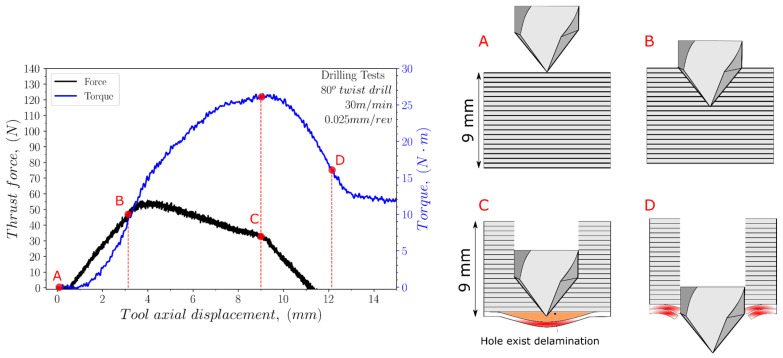
Evolution of both the thrust force and torque during the drilling of the UHMWPE composite with the 80° twist drill geometry (A–D letter correspond to different instants during the drilling operation).

**Figure 6 polymers-15-03882-f006:**
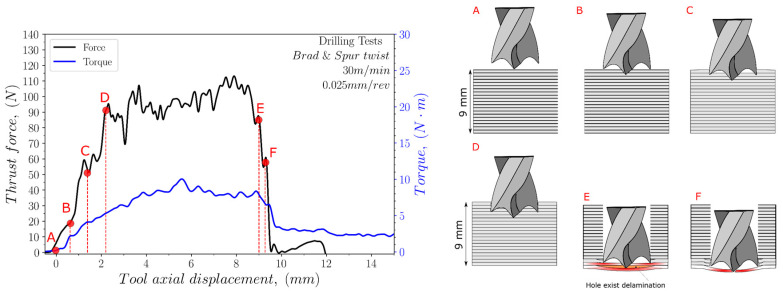
Evolution of both the thrust force and torque during the drilling of the UHMWPE composite with the brad and spur drill geometry (A–F letter correspond to different instants during the drilling operation).

**Figure 7 polymers-15-03882-f007:**
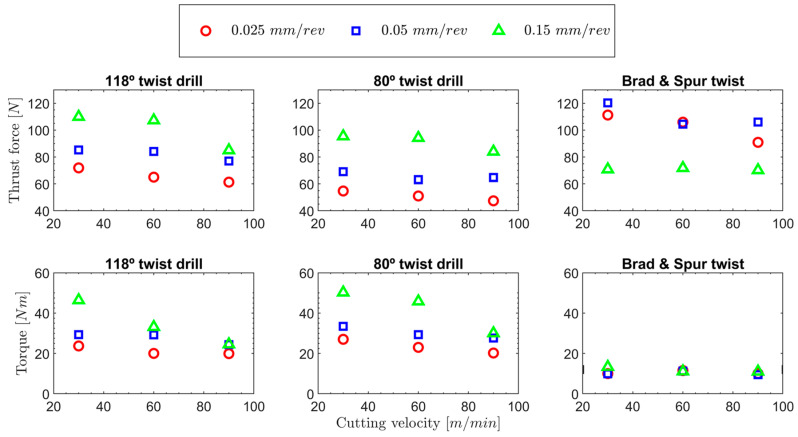
Thrust force and torque results obtained for the different tested drill geometries.

**Figure 8 polymers-15-03882-f008:**
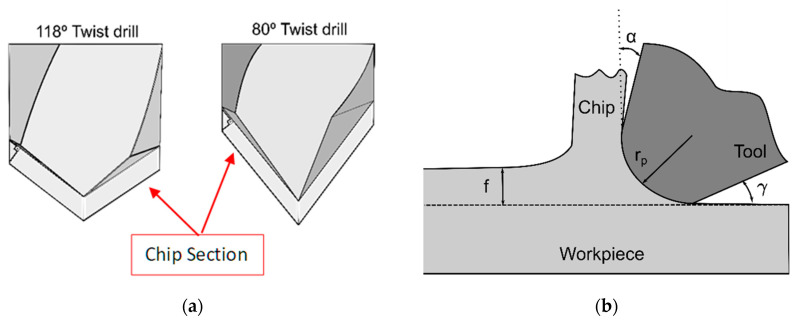
(**a**) Chip sections for both 118° and 80° twist drills; (**b**) effect of the cutting edge’s tip radius with respect to the feed.

**Figure 9 polymers-15-03882-f009:**
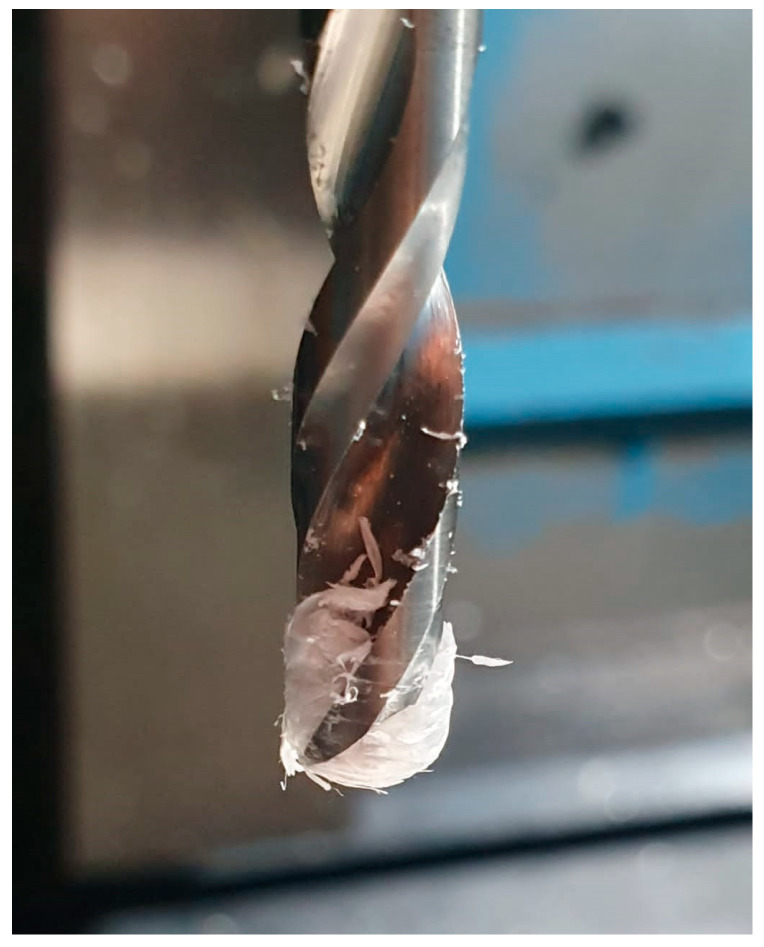
Plug generated at the tip of the bit during the drilling of the UHMWPE composite with the brad and spur geometry.

**Figure 10 polymers-15-03882-f010:**
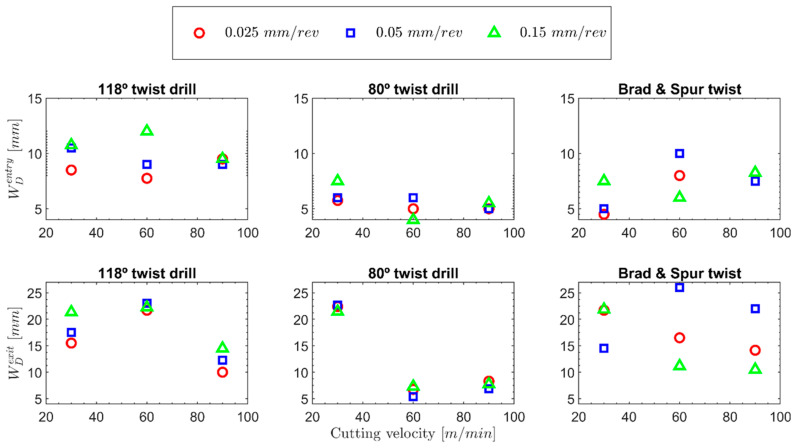
Delamination damage for all the tested cutting conditions and drill geometries.

**Figure 11 polymers-15-03882-f011:**
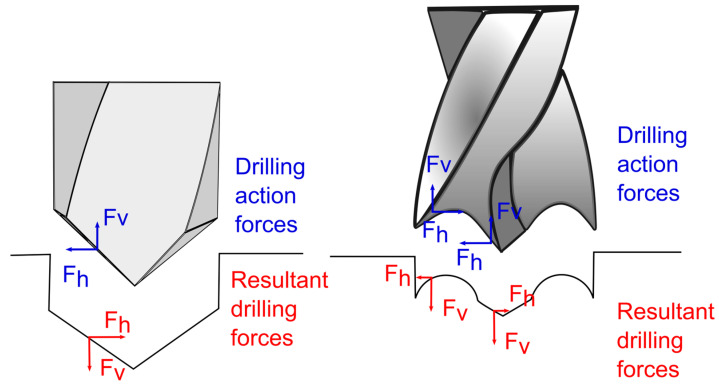
Resultant forces generated for the 118° twist drill and the brad and spur drill [2].

**Figure 12 polymers-15-03882-f012:**
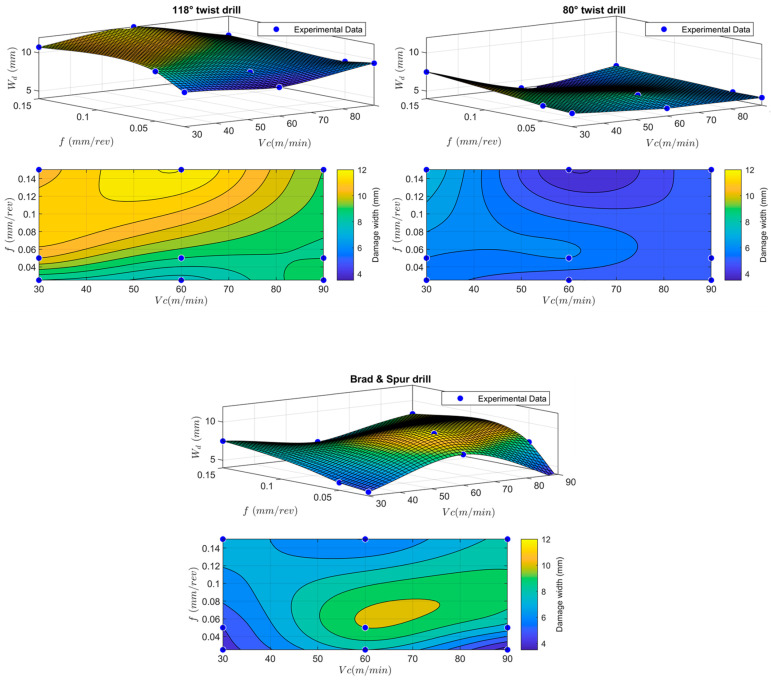
Three-dimensional (3D) maps of the delamination factor for the three studied drill geometries and cutting conditions.

**Figure 13 polymers-15-03882-f013:**
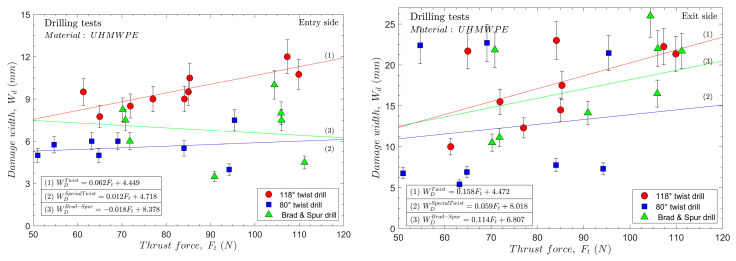
Delamination factor vs. thrust force.

**Figure 14 polymers-15-03882-f014:**
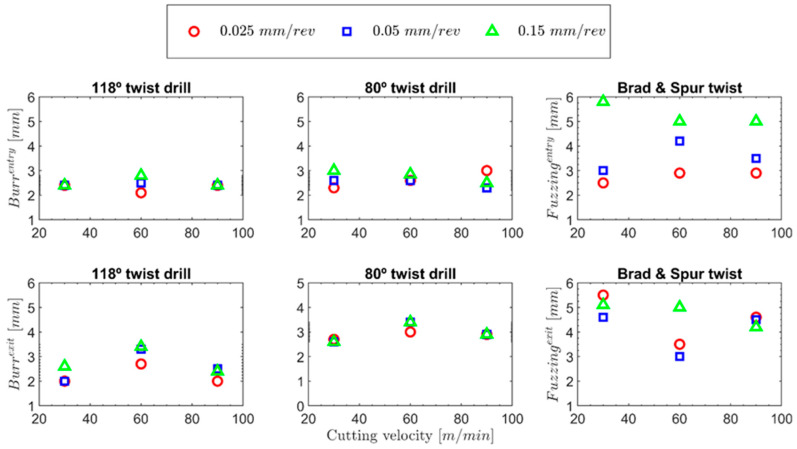
Burrs generated during drilling for both 118° and 80° twist drills and fuzzing generated for the brad and spur drill at the entrance and at the exit.

**Figure 15 polymers-15-03882-f015:**
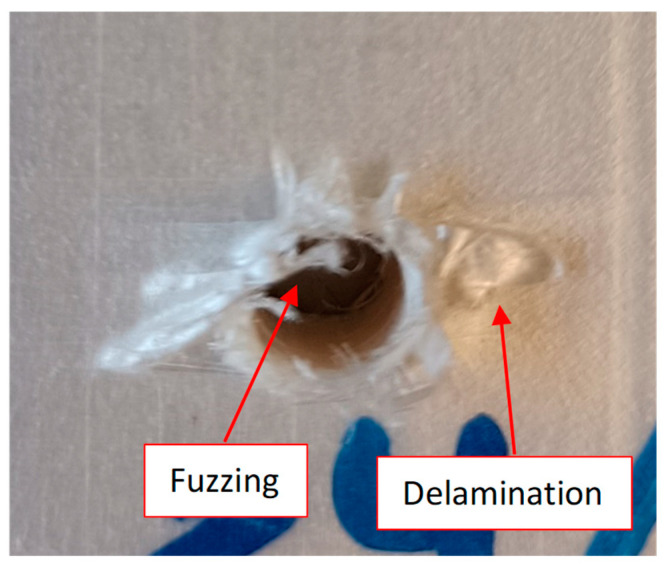
Delamination at entry during drilling of UHMWPE composite with the brad and spur drill at a cutting speed of 60 m/min and a feed of 0.15 mm/rev.

**Table 1 polymers-15-03882-t001:** Main properties of the UHMWPE composite [31].

Fibre Volume Fraction	Approx. 90%
Fibre Strength	1222 MPa
Fibre Modulus	130.6 GPa
Cross-Ply Thickness	50–60 µm
Configuration	(0°/90°/0°/90°)
Laminate Density	0.97 g/cm^3^

**Table 2 polymers-15-03882-t002:** Cutting conditions tested on UHMWPE composite.

	Vc [m/min]	f [mm/rev]
Cutting Conditions	30	0.025
	60	0.05
	90	0.15

**Table 3 polymers-15-03882-t003:** Delamination factors for the 118° twist drill and relative delamination factors with respect to the 118° twist drill, 80° twist drill, and brad and spur.

Vc [m/min]	f [mm/rev]	$W_{D}^{entry}$ Twist 118°	$W_{D}^{exit}$ Twist 118°	$\%\;W_{D}^{entry}$ Twist 80°	$\%\;W_{D}^{exit}$ Twist 80°	$\%\;W_{D}^{entry}$ Brad and Spur	$\%\;W_{D}^{exit}$ Brad and Spur
30	0.03	8.5	15.5	−32.4%	44.5%	−47.1%	40.0%
	0.05	10.5	17.5	−42.9%	29.7%	−52.4%	−17.1%
	0.15	10.8	21.4	−30.2%	0.5%	−30.2%	−1.6%
60	0.03	7.8	21.7	−35.5%	−68.9%	3.2%	−24.0%
	0.05	9.0	23.0	−33.3%	−76.5%	11.1%	13.0%
	0.15	12.0	22.3	−66.7%	−67.2%	−50.0%	−49.9%
90	0.03	9.5	10.0	−47.4%	−17.0%	−63.2%	41.5%
	0.05	9.0	12.3	−44.4%	−43.9%	−16.7%	78.9%
	0.15	9.5	14.5	−42.1%	−46.6%	−13.2%	−27.6%

**Table 4 polymers-15-03882-t004:** Pearson correlation analysis among drilling parameters, r: Pearson’s correlation coefficient. Strong correlation (>0.5) is in green, and medium correlations (0.3–0.49) are in grey.

Drilling Parameters	Quality Characteristics					
	Thrust Force	Torque	Burr Entry	Burr Exit	Delamination Entry	Delamination Exit
Cutting Speed	−0.23	−0.26	0	−0.037	−0.066	−0.52
Feed	0.23	0.39	0.44	0.14	0.23	−0.027

**Table 5 polymers-15-03882-t005:** Pearson correlation analysis among drilling parameters for each drill, r: Pearson’s correlation coefficient. Strong correlations (>0.5) are in green, and medium correlations (0.3–0.49) are in grey.

Drilling Parameters	Drill	Quality Characteristics					
		Thrust Force	Torque	Burr/Fuzzing Entry	Burr/Fuzzing Exit	Delamination Entry	Delamination Exit
Cutting Speed	118° twist	−0.422	−0.47	0	0.05	−0.13	−0.47
	80° twist	−0.26	−0.42	−0.11	0.49	−0.62	−0.53
	brad and spur	−0.37	−0.26	−0.26	−0.53	0.21	−0.26
Feed	118° twist	0.90	0.84	0.50	0.43	0.74	−0.027
	80° twist	0.95	0.90	0.19	0	0.16	0.26
	brad and spur	−0.63	0.26	0.96	0.08	0.40	−0.21

## Data Availability

The data that support the findings of this study are available on request from the corresponding author.

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
