# Peer review of "Drilling of Cross-Ply UHMWPE Laminates: A Study on the Effects of the Tool Geometry and Cutting Parameters on the Integrity of Components"

_polymers, 2023, doi:10.3390/polym15193882_

Round 1
Reviewer 1 Report
Dear Authors,
This research is interesting and important for understanding how tool geometry and cutting parameters affect the component's integrity. The experiment stated in the article involves the processing of a hot-pressed, widely used multiple-layer material. The correlation between cutting conditions, cutting forces, and the quality of the machined hole has been extensively investigated in this paper.
While this work is publishable, there are a number of issues that need to be addressed prior.
Specific Comments
1. Please include the numerical features of the improvements in lines 13–24.
2. It is required to add the necessary more details to the lines 159–170 in the section explaining the geometry of drills: rake angle, back angle, cutting shape, and chip flute characteristics. I'm thankful for it.
3. What cutting material was used in the work??
4. To support the use of a specific tool, it is advised to discuss the kinematic characteristics of the blade in the discussion of drill stresses in lines 226-231. I'm thankful for.
5. Line 331, clarify the way the not deformation thickness of chips that were was calculated. Nothing about it is plain from the description.
6. How were the burrs that controlled at the hole's entrance and exit managed?
7. Since that the drill's unique design for composite materials contains additional structural features on the periphery, it may be worthwhile to include the parameter of the main cutting edge's angle of inclination in the geometry analysis. In this aspect, it is incorrect to establish a direct comparison between composites processing drill designs and normal drill designs' point angles.
Author Response
The authors are extremely grateful to the reviewer for providing a very detailed summary of the improvements needed to the manuscript.
The authors accept the first version of the manuscript submitted could have been improved before it was submitted which would have made the reviewing task easier. That said the authors have now attended to all the comments, implemented the amendments suggested, and addressed each of the question, which has resulted in extensive improvements throughout the manuscript that is now being resubmitted for review.
To assist the reviewers with reviewing the amended manuscript, we have included a pdf formatted version with corrections and new content highlighted in yellow.
Below you can find in blue authors answer to reviewer comments.
This research is interesting and important for understanding how tool geometry and cutting parameters affect the component's integrity. The experiment stated in the article involves the processing of a hot-pressed, widely used multiple-layer material. The correlation between cutting conditions, cutting forces, and the quality of the machined hole has been extensively investigated in this paper.
While this work is publishable, there are a number of issues that need to be addressed prior.
Specific Comments:
R.- 1. Please include the numerical features of the improvements in lines 13–24.
A1.- New numerical features were included.
R2.- It is required to add the necessary more details to the lines 159–170 in the section explaining the geometry of drills: rake angle, back angle, cutting shape, and chip flute characteristics. I'm thankful for it.
A2.- Thanks for the comment. Images of the drill geometries have been added and improved with more information. However, for reasons of company confidentiality, only those that are accessible on its website have been published. No right to publish this information was obtained from the supplier, however the references to the tools used were included (so anyone can ask for this information to the supplier. In addition, it was explained that the tool with a smaller point angle was customized to get the same rake and clearance angle to those of the 118º point angle tool.
R3.- What cutting material was used in the work??
A3.- Thanks for your comment. The cutting material has been added in the tool description.
R4.- To support the use of a specific tool, it is advised to discuss the kinematic characteristics of the blade in the discussion of drill stresses in lines 226-231. I'm thankful for.
A4.- Thank you very much for your observation. We had not previously taken into account the aspect of stress distribution on the tool's cutting edge. Following the lines highlighted by the reviewer, additional explanatory lines have been added.
R5.- Line 331, clarify the way the not deformation thickness of chips that were was calculated. Nothing about it is plain from the description.
A5.- Dear reviewer, thank you for your comment, which assists us in enhancing the provided explanation. The chip thickness was not included since it is directly proportional to the feed. We aimed to emphasize the fact that when calculating the torque required to remove the material, the chip cross-sectional area is always referenced. A line was included to clarified that no computation was done to get the chip thickness.
R6.- How were the burrs that controlled at the hole's entrance and exit managed?.
A6.- Thank you for your comment, the following sentence has been modified and added to the paper to clarify its measurement.
“Burrs are measured through optical images of the holes. Using the plane that defines the surface of the test piece as a reference, the height of the burrs has been calculated at both the entrance and exit points.”.
R7.- Since that the drill's unique design for composite materials contains additional structural features on the periphery, it may be worthwhile to include the parameter of the main cutting edge's angle of inclination in the geometry analysis. In this aspect, it is incorrect to establish a direct comparison between composites processing drill designs and normal drill designs' point angles.
A7.-. Thank you very much for your valued feedback. To enhance the article, we have included greater detail of the tool in Figure 2, and we have also included the manufacturer's reference for the tool. Additionally, we have incorporated Table 3 to highlight that not only does the removal of burrs significantly reduce delamination at both the entry and exit points (under the optimal operating conditions for this tool, corresponding to the highest of the tested feed rates). The reason for including this specialized tool in the study was to investigate how it minimized burr formation and controlled delamination, which it did achieve. However, it was not the tool responsible for the most significant delamination reduction, and it also did not cut the fibers properly, rendering the hole unusable without post-processing.
Reviewer 2 Report
· As the main objective of the work done here is categorizing and defining the damage induced to the composite component during drilling, it is critical to define damage criteria for the machined structures as you explained on page 5. There are some good recently published works in terms of drilling quality assessment (pull-out and Peel-up) in composite laminates such as “Optimization of drilling parameters in composite sandwich structures (PVC core)”, “Experimental study of drilling behaviors and damage issues for woven GFRP composites using special drills” and “Defect evaluation of the honeycomb structures formed during the drilling process” and also more recently published papers in this area. I recommend you review and summarize them in the introduction section or the section “2.3. Damage mode” (may they have used different materials and help the readers to understand the challenge in materials machining).
· Page 3, line 145: “The laminate was manufactured using hot-pressing technology.”. It is necessary to add some details regarding the manufacturing parameters and conditions.
· Add appropriate references for Table 1.
· Do you think the drill diameter has a significant effect on the damage modes investigated in this research? Explain.
· Page 8, line 174: “For the development of the drilling test, three different cutting speeds and feeds are established following the main recommendations of the manufacturers”. What do mean by manufacturers? Manufactures of composite laminates? Or what?
· Add more details in the section “2.3. Damage mode” to clarify if you have investigated the damage modes in the entrance of the drill and also pull out damage of just damages in one side of the workpiece.
· Also, How did you monitor the damage areas on each side? Was it done visually or you have done NDT? Explain and clarify in the text.
Author Response
Reviewer 2
The authors are extremely grateful to the reviewer for providing a very detailed summary of the improvements needed to the manuscript.
The authors accept the first version of the manuscript submitted could have been improved before it was submitted which would have made the reviewing task easier. That said the authors have now attended to all the comments, implemented the amendments suggested, and addressed each of the question, which has resulted in extensive improvements throughout the manuscript that is now being resubmitted for review.
To assist the reviewers with reviewing the amended manuscript, we have included a pdf formatted version with corrections and new content highlighted in yellow.
Below you can find in blue authors answer to reviewer comments.
R1.- As the main objective of the work done here is categorizing and defining the damage induced to the composite component during drilling, it is critical to define damage criteria for the machined structures as you explained on page 5. There are some good recently published works in terms of drilling quality assessment (pull-out and Peel-up) in composite laminates such as “Optimization of drilling parameters in composite sandwich structures (PVC core)”, “Experimental study of drilling behaviors and damage issues for woven GFRP composites using special drills” and “Defect evaluation of the honeycomb structures formed during the drilling process” and also more recently published papers in this area. I recommend you review and summarize them in the introduction section or the section “2.3. Damage mode” (may they have used different materials and help the readers to understand the challenge in materials machining).
A1.- Thanks for your suggestion. New references and comments related to them have been added to the text.
“Ghabezi et al. on its works [38,39] already distinguished between burrs and delamination, highlighting two types of delamination based on the area where they appeared during the drilling of composite sandwich. Thus, he pointed out one damage fiber around the hole and the other mode is existing uncut fiber and matrix that is appeared into the hole.
Xu et al. [44] highlighted during the drilling of woven GFRP composites that the burrs phenomenon observed during machining is attributed mainly to the bouncing-back effects and mechanical recession of fibers under the bending loads of the tool lips.”
R2.- Page 3, line 145: “The laminate was manufactured using hot-pressing technology.”. It is necessary to add some details regarding the manufacturing parameters and conditions. Add appropriate references for Table 1.
A2.- The authors appreciate the reviewer's comment. Specific details of how it was manufactured in temperature and pressure cannot be provided due to confidentiality by the supplying company. However, a reference has been added that seems like the manufacturing conditions of the UHMPWE plate used in the present work. Add appropriate references for Table 1.
The following sentence has been added in the manuscript:
“following a procedure similar to that presented by Haris and Tan [31]”
Also the reference to the table was included.
[31] Haris, A.; Tan, V.B.C. Effects of spacing and ply blocking on the ballistic resistance of UHMWPE laminates. Int. J. Impact Eng. 2021, 151, 103824.
R3.- Do you think the drill diameter has a significant effect on the damage modes investigated in this research? Explain
A3.- In the study, the effect of diameter on delamination is not evaluated. What has been measured, however, is the relationship between feed force and delamination. Although we cannot make categorical statements based on the results obtained, it appears reasonable to expect that as the drill bit diameter increases, higher feed forces would be encountered for identical cutting parameters. Consequently, this would result in greater energy generated to break the cohesion between material layers, potentially leading to higher delaminations.
For thermoset composite materials, step drill bits may be used if the intention is to create large-diameter holes in a single pass (progressive drilling would be employed when multiple passes are feasible, gradually increasing the hole diameter). While there is limited research available for materials like the one included in our study, existing literature points in this direction. This consideration hasn't been included in the final paper because the test campaign was not oriented toward analyzing the diameter's effect on the observed delaminations, and we lacked data in the article to support this argument.
R4.- Page 8, line 174: “For the development of the drilling test, three different cutting speeds and feeds are established following the main recommendations of the manufacturers”. What do mean by manufacturers? Manufactures of composite laminates? Or what?
A4.- Thank you for your comment. Manufacturers in this case are related to tool manufacturers as well as their suppliers. All of them provide us with information regarding the best cutting parameters and tools. However, to better explain this point the following paragraph was included:
“For the development of the drilling test, three different cutting speeds and feeds are established considering a combination of factors, including the recommendations provided by tool manufacturers, insights from material suppliers, and findings reported in the literature by other authors.”
R5.- Add more details in the section “2.3. Damage mode” to clarify if you have investigated the damage modes in the entrance of the drill and also pull out damage of just damages in one side of the workpiece.
A5.- Thank you very much for your comment, the following sentence has been modified and added to the paper to clarify its measurement.
“Therefore, in the present study, both the delamination and the burrs generated during machining have been quantified at the entrance and at the exit of the hole.”
R6.- Also, How did you monitor the damage areas on each side? Was it done visually or you have done NDT? Explain and clarify in the text.
A6.- Thank you your comment, the following sentence has been modified and added to the paper to clarify its measurement.
“This damage has been determined visually by means of the calculation of the maximum diameter within the damage appears throughout the images obtained using an optical microscope Optika SZR.”
“Burrs are measured through optical images of the holes. Using the plane that defines the surface of the test piece as a reference, the height of the burrs has been calculated at both the entrance and exit points.”
Reviewer 3 Report
The authors are conducting a research study on drilling UHMWPE laminate, focusing on its relevance in the defense industry. They have prepared the material using hot-pressing technology and performed drilling experiments with various drill types. Data on cutting forces, failure modes, and damage types are collected and analyzed to assess machinability. The study aims to provide valuable insights and recommendations for machining UHMWPE laminate, crucial for applications in combat helmets and armor in the defense sector. The work is acceptable in my opinion but still requires improvement. I give some comments and questions so that the authors can improve their paper
1- What are the measured cutting forces when drilling UHMWPE laminate using HSS twist drills with different point angles, and how do these forces compare between the two drill types?
2- Could you provide the exact delamination percentages observed in the samples drilled with the brad & spur drill of 6 mm diameter, and how do these values compare to those obtained using HSS twist drills?
3- In terms of dimensional stability, how much does the UHMWPE laminate deviate from its specified dimensions after drilling with the different drill types, and are there significant differences in these deviations?
4- What are the specific values for the main damage modes observed in the UHMWPE laminate samples during drilling, and how do these modes vary between the different drills used?
5- Can you provide the precise recommendations for drill selection and machining conditions based on the cutting force performance values obtained in the study?
The English language in your article displays a commendable level of quality. Your sentences are skillfully constructed, successfully conveying the intended message.
Author Response
Reviewer 3
The authors are extremely grateful to the reviewer for providing a very detailed summary of the improvements needed to the manuscript.
The authors accept the first version of the manuscript submitted could have been improved before it was submitted which would have made the reviewing task easier. That said the authors have now attended to all the comments, implemented the amendments suggested, and addressed each of the question, which has resulted in extensive improvements throughout the manuscript that is now being resubmitted for review.
To assist the reviewers with reviewing the amended manuscript, we have included a pdf formatted version with corrections and new content highlighted in yellow.
Below you can find in blue authors answer to reviewer comments.
The authors are conducting a research study on drilling UHMWPE laminate, focusing on its relevance in the defense industry. They have prepared the material using hot-pressing technology and performed drilling experiments with various drill types. Data on cutting forces, failure modes, and damage types are collected and analyzed to assess machinability. The study aims to provide valuable insights and recommendations for machining UHMWPE laminate, crucial for applications in combat helmets and armor in the defense sector. The work is acceptable in my opinion but still requires improvement. I give some comments and questions so that the authors can improve their paper
R1.- What are the measured cutting forces when drilling UHMWPE laminate using HSS twist drills with different point angles, and how do these forces compare between the two drill types?
A1.- Dear reviewer, all cutting forces are included in Figure 7., in the text, we have included a general comparison between the force and cutting torque outputs for both geometries:
Furthermore, we have added the percentages relative to the variation found between both tools.
R2.- Could you provide the exact delamination percentages observed in the samples drilled with the brad & spur drill of 6 mm diameter, and how do these values compare to those obtained using HSS twist drills? What are the specific values for the main damage modes observed in the UHMWPE laminate samples during drilling, and how do these modes vary between the different drills used?
A2.- Dear reviewer, we sincerely appreciate your request for us to provide the precise values of delaminations and for giving us the opportunity to emphasize that this damage was consistent across all three tested tools. We have included Table 3, which presents the exact delamination values obtained with the reference tool (118° point angle tool) and the relative variations compared to the reference values for the tools with 80° point angle and Brad&spur. Furthermore, thanks to your feedback, we have been able to rectify Figure 13, where we identified an error in the Matlab code used to plot the results. We are deeply grateful for your assistance.
R3.- In terms of dimensional stability, how much does the UHMWPE laminate deviate from its specified dimensions after drilling with the different drill types, and are there significant differences in these deviations?
A3.- Dear reviewer, we sincerely appreciate your feedback. The authors have not included the mentioned study because conducting it would require post-processing the test specimens by removing the damages at the entrance and exit points, as the most significant deviations have been localized in these two areas. Removing these zones would result in a loss of critical information for future studies and could potentially introduce bias into these measurements. However, without the need for any specific measurements, the authors, through mere visual inspection and caliper usage, ascertain that the Brad&Spur drill bit exhibits the poorest control over hole dimensions due to observed fuzzing and material residue adhering to the tip under various conditions. Nevertheless, the differences in final hole dimensions between the drill bits with 80 and 118° point angles were found to be negligible.
R4.- Can you provide the precise recommendations for drill selection and machining conditions based on the cutting force performance values obtained in the study?
A4.- Thank you very much for your comment, the following sentence has been modified and added to the paper.
“• For more productive parameters, both the cutting speed and feed should be increased, being possible to verify that the drill that minimizes delamination is the one with the lowest point angle (80° twist drill), where a delamination of 7.75mm was found, the increase in delamination is 43.51% with respect to the minimum value obtained under optimal conditions. In terms of cutting force, again the drill with the lowest point angle (80° twist drill) obtained the best results when machining at the highest cutting speed and the lowest feed.”